# K for the Price of 1: Parameter-efficient Multi-task and Transfer Learning

**Pramod Kaushik Mudrarkarta**[*]
The University of Chicago
pramodkm@uchicago.edu

**Mark Sandler, Andrey Zhmoginov, Andrew Howard**
Google Inc.
{sandler,azhmogin,howarda}@google.com

## Abstract

We introduce a novel method that enables parameter-efficient transfer and multi-task learning with deep neural networks. The basic approach is to learn a model patch - a small set of parameters - that will specialize to each task, instead of fine-tuning the last layer or the entire network. For instance, we show that learning a set of scales and biases is sufficient to convert a pretrained network to perform well on qualitatively different problems (e.g. converting a Single Shot Multi-Box Detection (SSD) model into a 1000-class image classification model while reusing 98% of parameters of the SSD feature extractor). Similarly, we show that re-learning existing low-parameter layers (such as depth-wise convolutions) while keeping the rest of the network frozen also improves transfer-learning accuracy significantly. Our approach allows both simultaneous (multi-task) as well as sequential transfer learning. In several multi-task learning problems, despite using much fewer parameters than traditional logits-only fine-tuning, we match single-task performance.

## 1 Introduction

Deep neural networks have revolutionized many areas of machine intelligence and are now used for many vision tasks that even few years ago were considered nearly impenetrable (Krizhevsky et al., 2012; Simonyan & Zisserman, 2014; Liu et al., 2016). Advances in neural networks and hardware is resulting in much of the computation being shifted to consumer devices, delivering faster response, and better security and privacy guarantees (Konečný et al., 2016; Howard et al., 2017).

As the space of deep learning applications expands and starts to personalize, there is a growing need for the ability to quickly build and customize models. While model sizes have dropped dramatically from >50M parameters of the pioneering work of AlexNet (Krizhevsky et al., 2012) and VGG (Simonyan & Zisserman, 2014) to <5M of the recent Mobilenet (Sandler et al., 2018; Howard et al., 2017) and ShuffleNet (Zhang et al., 2017; Ma et al., 2018), the accuracy of models has been improving. However, delivering, maintaining and updating hundreds of models on the embedded device is still a significant expense in terms of bandwidth, energy and storage costs.

While there still might be space for improvement in designing smaller models, in this paper we explore a different angle: we would like to be able to build models that require only a few parameters to be trained in order to be re-purposed to a different task, with minimal loss in accuracy compared to a model trained from scratch. While there is ample existing work on compressing models and learning as few weights as possible (Rosenfeld & Tsotsos, 2018; Sandler et al., 2018; Howard et al., 2017) to solve a single task, to the best of our awareness, there is no prior work that tries to minimize the number of model parameters when solving many tasks together.

Our contribution is a novel learning paradigm in which each task carries its own **model patch** – a small set of parameters – that, along with a shared set of parameters constitutes the model for that task (for a visual description of the idea, see Figure 1, left side). We put this idea to use in two scenarios: a) in transfer learning, by fine-tuning only the model patch for new tasks, and b) in multi-task learning, where each task performs gradient updates to both its own model patch, and the

---

[*]Work done while at Google.

shared parameters. In our experiments (Section 5), the largest patch that we used is smaller than 10% of the size of the entire model. We now describe our contribution in detail.

**Transfer learning** We demonstrate that, by fine-tuning less than 35K parameters in MobilenetV2 (Sandler et al., 2018) and InceptionV3 (Szegedy et al., 2016), our method leads to significant accuracy improvements over fine-tuning only the last layer (102K-1.2M parameters, depending on the number of classes) on multiple transfer learning tasks. When combined with fine-tuning the last layer, we train less than 10% of the model's parameters in total.We also show the effectiveness of our method over last-layer-based fine-tuning on transfer learning between completely different problems, namely COCO-trained SSD model (Liu et al., 2016) to classification over ImageNet (Deng et al., 2009).

**Multi-task learning** We explore a multi-task learning paradigm wherein multiple models that share most of the parameters are trained simultaneously (see Figure 1, right side). Each model has a task-specific model patch. Training is done in a distributed manner; each task is assigned a subset of available workers that send independent gradient updates to both shared and task-specific parameters using standard optimization algorithms. Our results show that simultaneously training two such MobilenetV2 (Sandler et al., 2018) models on ImageNet (Deng et al., 2009) and Places-365 (Zhou et al., 2017) reach accuracies comparable to, and sometimes higher than individually trained models.

**Domain adaptation** We apply our multi-task learning paradigm to domain adaptation. For ImageNet (Deng et al., 2009), we show that we can simultaneously train MobilenetV2 (Sandler et al., 2018) models operating at 5 different resolution scales, 224, 192, 160, 128 and 96, while sharing more than 98% of the parameters and resulting in the same or higher accuracy as individually trained models. This has direct practical benefit in power-constrained operation, where an application can switch to a lower resolution to save on latency/power, without needing to ship separate models and having to make that trade-off decision at the application design time. The cascade algorithm from Streeter (2018) can further be used to reduce the average running time by about 15% without loss in accuracy.

The rest of the paper is organized as follows: we describe our method in Section 2 and discuss related work in Section 3. In Section 4, we present simple mathematical intuition that contrasts the expressiveness of logit-only fine-tuning and that of our method. Finally, in Section 5, we present detailed experimental results.

## 2 METHOD

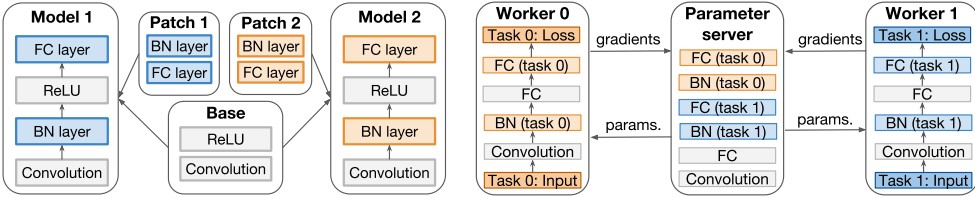

Figure 1: Left: An example illustrating the idea of a model patch. Right: An example of multi-task learning 2. FN: fully connected layer, BN: batch normalization layer, and Conv: convolution layer.

The central concept in our method is that of a **model patch**. It is essentially a small set of per-channel transformations that are dispersed throughout the network resulting in only a tiny increase in the number of model parameters.

Suppose a deep network $\mathcal{M}$ is a sequence of layers represented by their parameters (weights, biases), $\mathbf{W}_1, \ldots, \mathbf{W}_n$. We ignore non-trainable layers (e.g., some kinds of activations) in this formulation. A model patch $P$ is a set of parameters $\mathbf{W}'_{i_1}, \ldots, \mathbf{W}'_{i_k}, 1 \leq i_1, \ldots, i_k \leq n$ that, when applied to $\mathcal{M}$, adds layers at positions $i_1, \ldots, i_n$. Thus, a patched model

$$\mathcal{M}' = \mathbf{W}_1, \ldots, \mathbf{W}_{i_1}, \mathbf{W}'_{i_1}, \ldots, \mathbf{W}_{i_n}, \mathbf{W}'_{i_n}, \ldots, \mathbf{W}_n$$

In this paper, we introduce two kinds of patches. We will see below that they can be folded with the other layers in the network, eliminating the need to perform any explicit addition of layers. In Section 5, we shed some light on why the particular choice of these patches is important.

**Scale-and-bias patch**    This patch applies per-channel scale and bias to every layer in the network. In practice this transformations can often be absorbed into normalization layer such as Batch Normalization (Ioffe & Szegedy, 2015). Let $\mathbf{X}$ be an activation tensor. Then, the batch-normalized version of $\mathbf{X}$

$$BN(\mathbf{X}) = \gamma \frac{\mathbf{X} - \mu(\mathbf{X})}{\sigma(\mathbf{X})} + \beta$$

where $\mu(\mathbf{X}), \sigma(\mathbf{X})$ are mean and standard deviation computed per minibatch, and $\gamma, \beta$ are learned via backpropagation. These statistics are computed as mini-batch average, while during inference they are computed using global averages.

The scale-and-bias patch corresponds to all the $\gamma, \beta, \mu, \sigma$ in the network. Using BN as the model patch also satisfies the criterion that the patch size should be small. For instance, the BN parameters in both MobilenetV2 (Sandler et al., 2018) and InceptionV3 network performing classification on ImageNet amounts to less than 40K parameters, of about 1% for MobilenetV2 that has 3.5 million Parameters, and less than 0.2% for Inception V3 that has 25 million parameters.

While we utilize batch normalization in this paper, we note that this is merely an implementation detail and we can use explicit biases and scales with similar results.

**Depthwise-convolution patch**    The purpose of this patch is to re-learn spatial convolution filters in a network. Depth-wise separable convolutions were introduced in deep neural networks as way to reduce number of parameters without losing much accuracy (Howard et al., 2017; Chollet, 2017). They were further developed in Sandler et al. (2018) by adding linear bottlenecks and expansions.

In depthwise separable convolutions, a standard convolution is decomposed into two layers: a depthwise convolution layer, that applies one convolutional filter per input channel, and a pointwise layer that computes the final convolutional features by linearly combining the depthwise convolutional layers' output across channels. We find that the set of depthwise convolution layers can be re-purposed as a model patch. They are also lightweight - for instance, they account for less than 3% of MobilenetV2's parameters when training on ImageNet.

Next, we describe how model patches can be used in transfer and multi-task learning.

**Transfer learning**    In transfer learning, the task is to adapt a pretrained model to a new task. Since the output space of the new task is different, it necessitates re-learning the last layer. Following our approach, we apply a model patch and train the patched parameters, optionally also the last layer. The rest of the parameters are left unchanged. In Section 5, we discuss the inclusion/exclusion of the last layer. When the last layer is not trained, it is fixed to its random initial value.

**Multitask learning**    We aim to simultaneously, but independently, train multiple neural networks that share most weights. Unlike in transfer learning, where a large fraction of the weights are kept frozen, here we learn all the weights. However, each task carries its own model patch, and trains a patched model. By training all the parameters, this setting offers more adaptability to tasks while not compromising on the total number of parameters.

To implement multi-task learning, we use the distributed TensorFlow paradigm[1]: a central parameter server receives gradient updates from each of the workers and updates the weights. Each worker reads the input, computes the loss and sends gradients to the parameter server. We allow subsets of workers to train different tasks; workers thus may have different computational graphs, and task-specific input pipelines and loss functions. A visual depiction of this setting is shown in Figure 1.

---

[1] `https://www.tensorflow.org/guide/extend/architecture`

## 3 RELATED WORK

One family of approaches (Yosinski et al., 2014; Donahue et al., 2014) widely used by practitioners for domain adaptation and transfer learning is based on fine-tuning only the last layer (or sometimes several last layers) of a neural network to solve a new task. Fine-tuning the last layer is equivalent to training a linear classifier on top of existing features. This is typically done by running SGD while keeping the rest of the network fixed, however other methods such as SVM has been explored as well (Kim et al., 2013). It has been repeatedly shown that this approach often works best for similar tasks (for example, see Donahue et al. (2014)).

Another frequently used approach is to use full fine-tuning (Cui et al., 2018) where a pretrained model is simply used as a warm start for the training process. While this often leads to significantly improved accuracy over last-layer fine-tuning, downsides are that 1) it requires one to create and store a full model for each new task, and 2) it may lead to overfitting when there is limited data. In this work, we are primarily interested in approaches that allow one to produce highly accurate models while reusing a large fraction of the weights of the original model, which also addresses the overfitting issue.

While the core idea of our method is based on learning small model patches, we see significant boost in performance when we fine-tune the patch along with last layer (Section 5). This result is somewhat in contrast with Hoffer et al. (2018), where the authors show that the linear classifier (last layer) does not matter when training full networks. Mapping out the conditions of when a linear classifier can be replaced with a random embedding is an important open question.

Li et al. (2016) show that re-computing batch normalization statistics for different domains helps to improve accuracy. In Rosenfeld & Tsotsos (2018) it was suggested that learning batch normalization layers in an otherwise randomly initialized network is sufficient to build non-trivial models. Re-computing batch normalization statistics is also frequently used for model quantization where it prevents the model activation space from drifting (Krishnamoorthi, 2018). In the present work, we significantly broaden and unify the scope of the idea and scale up the approach by performing transfer and multi-task learning across completely different tasks, providing a powerful tool for many practical applications.

Our work has interesting connections to meta-learning (Nichol & Schulman, 2018; Finn et al., 2017; Chen et al., 2018). For instance, when training data is not small, one can allow each task to carry a small model patch in the Reptile algorithm of Nichol & Schulman (2018) in order to increase expressivity at low cost.

## 4 ANALYSIS

Experiments (Section 5) show that model-patch based fine-tuning, especially with the scale-and-bias patch, is comparable and sometimes better than last-layer-based fine-tuning, despite utilizing a significantly smaller set of parameters. At a high level, our intuition is based on the observation that individual channels of hidden layers of neural network form an embedding space, rather than correspond to high-level features. Therefore, even simple transformations to the space could result in significant changes in the target classification of the network.

In this section (and in Appendix A), we attempt to gain some insight into this phenomenon by taking a closer look at the properties of the last layer and studying low-dimensional models.

A deep neural network performing classification can be understood as two parts:

1. a network base corresponding to a function $F : \mathbb{R}^d \to \mathbb{R}^n$ mapping $d$-dimensional input space $\mathbb{X}$ into an $n$-dimensional embedding space $\mathbb{G}$, and

2. a linear transformation $s : \mathbb{R}^n \to \mathbb{R}^k$ mapping embeddings to logits with each output component corresponding to an individual class.

An input $\boldsymbol{x} \in \mathbb{X}$ producing the output $\boldsymbol{o} := s(F(\boldsymbol{x})) \in \mathbb{R}^k$ is assigned class $c$ iff $\forall i \neq c, o_i < o_c$.

We compare fine-tuning model patches with fine-tuning only the final layer $s$. Fine-tuning only the last layer has a severe limitation caused by the fact that linear transformations preserve convexity.

It is easy to see that, regardless of the details of $s$, the mapping from embeddings to logits is such that if both $\boldsymbol{\xi}^a, \boldsymbol{\xi}^b \in \mathbb{G}$ are assigned label $c$, the same label is assigned to every $\boldsymbol{\xi}^\tau := \tau \boldsymbol{\xi}^b + (1 - \tau)\boldsymbol{\xi}^a$ for $0 \le \tau \le 1$. Indeed, $[s(\boldsymbol{\xi}^\tau)]_c = \tau o_c^b + (1 - \tau)o_c^a > \tau o_i^b + (1 - \tau)o_i^a = [s(\boldsymbol{\xi}^\tau)]_i$ for any $i \ne c$ and $0 \le \tau \le 1$, where $\boldsymbol{o}^a := s(\boldsymbol{\xi}^a)$ and $\boldsymbol{o}^b := s(\boldsymbol{\xi}^b)$. Thus, if the model assigns inputs $\{\boldsymbol{x}_i | i = 1, \ldots, n_c\}$ some class $c$, then the same class will also be assigned to any point in the preimage of the convex hull of $\{F(\boldsymbol{x}_i) | i = 1, \ldots, n_c\}$.

This property of the linear transformation $s$ limits one's capability to tune the model given a new input space manifold. For instance, if the input space is "folded" by $F$ and the neighborhoods of very different areas of the input space $\mathbb{X}$ are mapped to roughly the same neighborhood of the embedding space, the final layer cannot disentangle them while operating on the embedding space alone (should some new task require differentiating between such "folded" regions).

We illustrate the difference in expressivity between model-patch-based fine-tuning and last-layer-based fine-tuning in the cases of 1D (below) and 2D (Appendix A) inputs and outputs. Despite the simplicity, our analysis provides useful insights into how by simply adjusting biases and scales of a neural network, one can change which regions of the input space are folded and ultimately the learned classification function.

In what follows, we will work with a construct introduced by Montufar et al. (2014) that demonstrates how neural networks can "fold" the input space $\mathbb{X}$ a number of times that grows exponentially with the neural network depth[2]. We consider a simple neural network with one-dimensional inputs and outputs and demonstrate that a single bias can be sufficient to alter the number of "folds", the topology of the $\mathbb{X} \to \mathbb{G}$ mapping. More specifically, we illustrate how the number of connected components in the preimage of a one-dimensional segment $[\boldsymbol{\xi}^a, \boldsymbol{\xi}^b]$ can vary depending on a value of a single bias variable.

As in Montufar et al. (2014), consider the following function:

$$q(x; b) \equiv 2 \operatorname{ReLU}\left([1, -1, \ldots, (-1)^{p-1}] \cdot \boldsymbol{v}^T(x; \boldsymbol{b}) + b_p\right),$$

where

$$\boldsymbol{v}(x; \boldsymbol{b}) \equiv [\max(0, x + b_0), \max(0, 2x - 1 + b_1), \ldots, \max(0, 2x - (p - 1) + b_{p-1})],$$

$p$ is an even number, and $\boldsymbol{b} = (b_0, \ldots, b_p)$ is a $(p + 1)$–dimensional vector of tunable parameters characterizing $q$. Function $q(x; \boldsymbol{b})$ can be represented as a two-layer neural network with ReLU activations.

Set $p = 2$. Then, this network has 2 hidden units and a single output value, and is capable of "folding" the input space twice. Defining $F$ to be a composition of $k$ such functions

$$F(x; \boldsymbol{b}^{(1)}, \ldots, \boldsymbol{b}^{(k)}) \equiv q(q(\ldots q(q(x; \boldsymbol{b}^{(1)}); \boldsymbol{b}^{(2)}); \ldots; \boldsymbol{b}^{(k-1)}); \boldsymbol{b}^{(k)}), \qquad (1)$$

we construct a neural network with $2k$ layers that can fold input domain $\mathbb{R}$ up to $2^k$ times. By plotting $F(x)$ for $k = 2$ and different values of $b_0^{(1)}$ while fixing all other biases to be zero, it is easy to observe that the preimage of a segment $[0.2, 0.4]$ transitions through several stages (see figure 2), in which it: (a) first contains 4 disconnected components for $b_0^{(1)} > -0.05$, (b) then 3 for $b_0^{(1)} \in (-0.1, -0.05]$, (c) 2 for $b_0^{(1)} \in (-0.4, -0.1]$, (d) becomes a simply connected segment for $b_0^{(1)} \in [-0.45, -0.4]$ and (e) finally becomes empty when $b_0^{(1)} < -0.45$. This result can also be extended to $k > 2$, where, by tuning $b_0^{(1)}$, the number of "folds" can vary from $2^k$ to 0.

## 5 EXPERIMENTS

We evaluate the performance of our method in both transfer and multi-task learning using the image recognition networks MobilenetV2 (Sandler et al., 2018) and InceptionV3 (Szegedy et al., 2016) and a variety of datasets: ImageNet (Deng et al., 2009), CIFAR-10/100 (Krizhevsky, 2009), Cars (Krause

---

[2]In other words, $F^{-1}(\boldsymbol{\xi})$ for some $\boldsymbol{\xi}$ contains an exponentially large number of disconnected components.

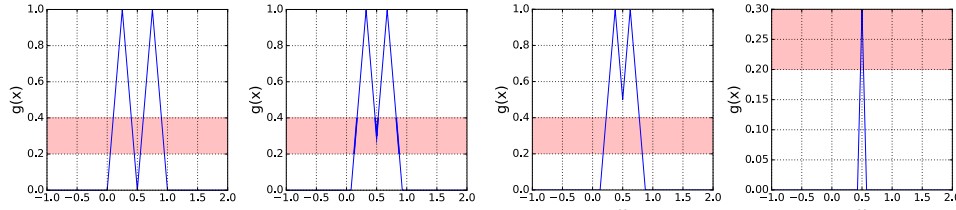

Figure 2: Function plots $F(x; \boldsymbol{b}^{(1)}, \boldsymbol{b}^{(2)})$ for a 4-layer network given by equation 1 with $k = 2$ and all biases except $b_0^{(1)}$ set to zero. From left to right: $b_0^{(1)} = 0$, $b_0^{(1)} = -0.075$, $b_0^{(1)} = -0.125$ and $b_0^{(1)} = -0.425$. The preimage of a segment $[0.2, 0.4]$ (shown as shaded region) contains 4, 3, 2 and 1 connected components respectively.

et al., 2013), Aircraft (Maji et al., 2013), Flowers-102 (Nilsback & Zisserman, 2008) and Places-365 (Zhou et al., 2017). An overview of these datasets can be found in Table 1. We also show preliminary results on transfer learning across completely different types of tasks using MobilenetV2 and Single-Shot Multibox Detector (SSD) (Liu et al., 2016) networks.

We use both scale-and-bias (S/B) and depthwise-convolution patches (DW) in our experiments. Both MobilenetV2 and InceptionV3 have batch normalization - we use those parameters as the S/B patch. MobilenetV2 has depthwise-convolutions from which we construct the DW patch. In our experiments, we also explore the effect of fine-tuning the patches along with the last layer of the network. We compare with two scenarios: 1) only fine-tuning the last layer, and 2) fine-tuning the entire network.

Table 1: Datasets used in experiments (Section 5)

| Name | CIFAR-100 | Flowers-102 | Cars | Aircraft | Places-365 | ImageNet |
|---|---|---|---|---|---|---|
| #images | 60,000 | 8,189 | 16,185 | 10,200 | 1.8 million | 1.3 million |
| #classes | 100 | 102 | 196 | 102 | 365 | 1000 |

We use TensorFlow (Abadi et al., 2015), and NVIDIA P100 and V100 GPUs for our experiments. Following the standard setup of Mobilenet and Inception we use $224 \times 224$ images for MobilenetV2 and $299 \times 299$ for InceptionV3. As a special-case, for Places-365 dataset, we use $256 \times 256$ images. We use RMSProp optimizer with a learning rate of 0.045 and decay factor 0.98 per 2.5 epochs.

### 5.1 LEARNING WITH RANDOM WEIGHTS

To demonstrate the expressivity of the biases and scales, we perform an experiment on MobilenetV2, where we learn only the scale-and-bias patch while keeping the rest of the parameters frozen at their initial random state. The results are shown in Table 3 (right side). It is quite striking that simply adjusting biases and scales of random embeddings provides features powerful enough that even a linear classifier can achieve a non-trivial accuracy. Furthermore, the synergy exhibited by the combination of the last layer and the scale-and-bias patch is remarkable.

### 5.2 TRANSFER LEARNING

We take MobileNetV2 and InceptionV3 models pretrained on ImageNet (Top1 accuracies 71.8% and 76.6% respective), and fine-tune various model patches for other datasets.

Results on InceptionV3 are shown in Table 2. We see that fine-tuning only the scale-and-bias patch (using a fixed, random last layer) results in comparable accuracies as fine-tuning only the last layer while using fewer parameters. Compared to full fine-tuning (Cui et al., 2018), we use orders of magnitude fewer parameters while achieving nontrivial performance. Our results using MobilenetV2 are similar (more on this later).

In the next experiment, we do transfer learning between completely different tasks. We take an 18-category object detection (SSD) model (Liu et al., 2016) pretrained on COCO images (Lin et al.,

Table 2: Transfer-learning on Inception V3, against full-network fine-tuning.

| Fine-tuned params. | Flowers | | Cars | | Aircraft | |
|---|---|---|---|---|---|---|
| | Acc. | #params | Acc. | #params | Acc. | #params |
| Last layer | 84.5 | 208K | 55 | 402K | 45.9 | 205K |
| S/B + last layer | **90.4** | 244K | **81** | 437K | **70.7** | 241K |
| S/B only (random last) | 79.5 | **36K** | 33 | **36K** | 52.3 | **36K** |
| All (ours) | 93.3 | 25M | 92.3 | 25M | 87.3 | 25M |
| All (Cui et al., 2018) | 96.3 | 25M | 91.3 | 25M | 82.6 | 25M |

2014) and repurpose it for image classification on ImageNet. The SSD model uses MobilenetV2 (minus the last layer) as a featurizer for the input image. We extract it, append a linear layer and then fine-tune. The results are shown in Table 3. Again, we see the effectiveness of training the model patch along with the last layer - a 2% increase in the parameters translates to 19.4% increase in accuracy.

Table 3: Learning Imagenet from SSD feature extractor (left) and random filters (right)

| Fine-tuned params. | #params | COCO→Imagenet, Top1 | Random→ Imagenet, Top1 |
|---|---|---|---|
| Last layer | 1.31M | 29.2% | 0% |
| S/B + last layer | 1.35M | 47.8% | 20% |
| S/B only | 34K | 6.4% | 2.3% |
| All params | 3.5M | 71.8% | 71.8% |

Next, we discuss the effect of learning rate. It is common practice to use a small learning rate when fine-tuning the entire network. The intuition is that, when all parameters are trained, a large learning rate results in network essentially forgetting its initial starting point. Therefore, the choice of learning rate is a crucial factor in the performance of transfer learning. In our experiments (Appendix B.2, Figure 9) we observed the opposite behavior when fine-tuning only small model patches: the accuracy grows as learning rate increases. In practice, fine-tuning a patch that includes the last layer is more stable w.r.t. the learning rate than full fine-tuning or fine-tuning only the scale-and-bias patch.

Finally, an overview of results on MobilenetV2 with different learning rates and model patches is shown in Figure 3. The effectiveness of small model patches over fine-tuning only the last layer is again clear. Combining model patches and fine-tuning results in a synergistic effect. In Appendix B, we show additional experiments comparing the importance of learning custom bias/scale with simply updating batch-norm statistics (as suggested by Li et al. (2016)).

## 5.3 MULTI-TASK LEARNING

In this section we show that, when using model-specific patches during multi-task training, it leads to performance comparable to that of independently trained models, while essentially using a single model.

We simultaneously train MobilenetV2 (Sandler et al., 2018) on two large datasets: ImageNet and Places365. Although the network architecture is the same for both datasets, each model has its own private patch that, along with the rest of the model weights constitutes the model for that dataset. We choose a combination of the scale-and-bias patch, and the last layer as the private model patch in this experiment. The rest of the weights are shared and receive gradient updates from all tasks.

In order to inhibit one task from dominating the learning of the weights, we ensure that the learning rates for different tasks are comparable at any given point in time. This is achieved by setting hyperparameters such that the ratio of dataset size and the number of epochs per learning rate decay step is the same for all tasks. We assign the same number of workers for each task in the distributed learning environment. The results are shown in Table 4.

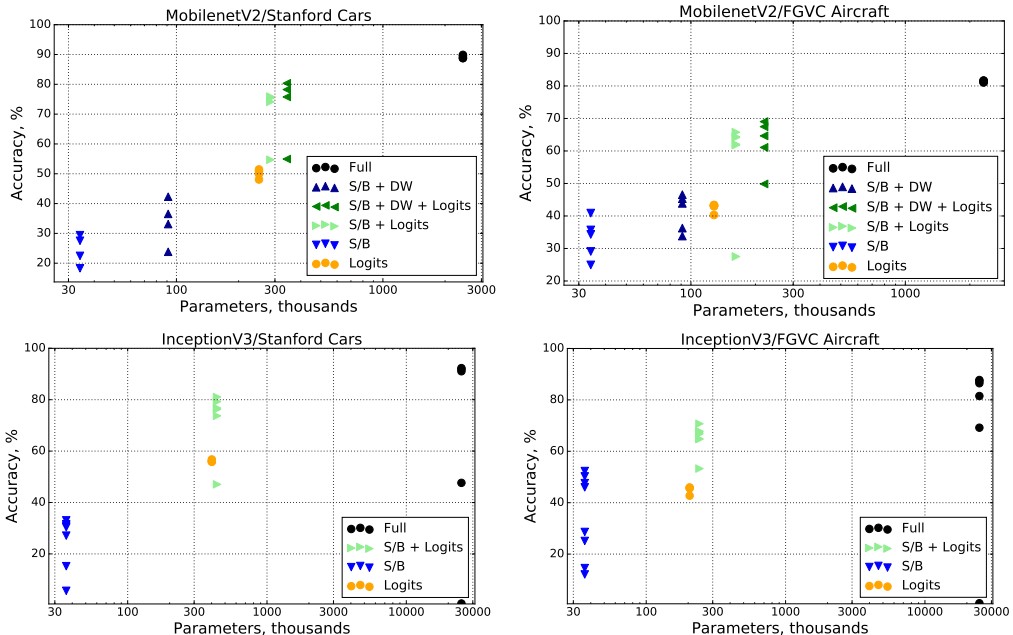

Figure 3: Performance of different fine-tuning approaches for different datasets for Mobilenet V2 and Inception. The same color points correspond to runs with different initial learning rates, starting from 0.0045 to 0.45 with factor 3. Best viewed in color.

Table 4: Multi-task learning with MobilenetV2 on ImageNet and Places-365.

| Task | S/B patch + last layer | Last layer | Independently trained |
|---|---|---|---|
| Imagenet | 70.2% | 64.4% | 71.8% |
| Places365 | **54.3**% | 51.4% | 54.2% |
| # total parameters | 3.97M | 3.93M | 6.05M |

Multi-task validation accuracy using a separate S/B patch for each model, is comparable to single-task accuracy, while considerably better than the setup that only uses separate logit-layer for each task, while using only using 1% more parameters (and 50% less than the independently trained setup).

## 5.4 DOMAIN ADAPTATION

In this experiment, each task corresponds to performing classification of ImageNet images at a different resolution. This problem is of great practical importance because it allows one to build very compact set of models that can operate at different speeds that can be chosen at inference time depending on power and latency requirements. Unlike in Section 5.3, we only have the scale-and-bias patch private to each task; the last layer weights are shared. We use bilinear interpolation to scale images before feeding them to the model. The learning rate schedule is the same as in Section 5.3.

The results are shown in Table 5. We compare our approach with S/B patch only against two baseline setups. *All shared* is where all parameters are shared across all models and *individually trained* is a much more expensive setup where each resolution has its own model. As can be seen from the table, scale-and-bias patch allows to close the accuracy gap between these two setups and even leads to a slight increase of accuracy for a couple of the models at the cost of 1% of extra parameters per each resolution.

Table 5: Multi-task accuracies of 5 MobilenetV2 models acting at different resolutions on ImageNet.

| Image resolution | S/B patch | All shared | Independently trained |
|---|---|---|---|
| 96 x 96 | 60.3% | 52.6% | 60.3% |
| 128 x 128 | **66.3%** | 62.4% | 65.3% |
| 160 x 160 | **69.5%** | 67.2% | 68.8% |
| 192 x 192 | 71% | 69.4% | 70.7% |
| 224 x 224 | 71.8% | 70.6% | 71.8% |
| # total parameters | **3.7M** | 3.5M | 17.7M |

## 6 CONCLUSIONS, OPEN QUESTIONS AND FUTURE WORK

We introduced a new way of performing transfer and multi-task learning where we patch only a very small fraction of model parameters, that leads to high accuracy on very different tasks, compared to traditional methods. This enables practitioners to build a large number of models with small incremental cost per model. We have demonstrated that using biases and scales alone allows pretrained neural networks to solve very different problems. While we see that model patches can adapt to a fixed, random last layer (also noted in Hoffer et al. (2018)), we see a significant accuracy boost when we allow the last layer also to be trained. It is important to close this gap in our understanding of when the linear classifier is important for the final performance. From an analytical perspective, while we demonstrated that biases alone maintain high expressiveness, more rigorous analysis that would allow us to predict which parameters are important, is still a subject of future work. From practical perspective, cross-domain multi-task learning (such as segmentation and classification) is a promising direction to pursue. Finally our approach provides for an interesting extension to the federated learning approach proposed in Konečný et al. (2016), where individual devices ship their gradient updates to the central server. In this extension we envision user devices keeping their local private patch to maintain personalized model while sending common updates to the server.

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

## A  ANALYSIS: 2D CASE

Here we show an example of a simple network that "folds" input space in the process of training and associates identical embeddings to different points of the input space. As a result, fine-tuning the final linear layer is shown to be insufficient to perform transfer learning to a new dataset. We also show that the same network can learn alternative embedding that avoids input space folding and permits transfer learning.

Consider a deep neural network mapping a 2D input into 2D logits via a set of 5 ReLU hidden layers: 2D input → 8D state → 16D state → 16D state → 8D state → $m$-D embedding (no ReLU) → 2D logits (no ReLU). Since the embedding dimension is typically smaller than the input space dimension, but larger than the number of categories, we first choose the embedding dimension $m$ to be 2. This network is trained (applying sigmoid to the logits and using cross entropy loss function) to map $(x, y)$ pairs to two classes according to the groundtruth dependence depicted in figure 4(a). Learned function is shown in figure 4(c). The model is then fine-tuned to approximate categories shown in figure 4(b). Fine-tuning all variables, the model can perfectly fit this new data as shown in figure 4(d).

Once the set of trainable parameters is restricted, model fine-tuning becomes less efficient. Figures 4(A) through 4(E) show output values obtained after fine-tuning different sets of parameters. In particular, it appears that training the last layer alone (see figure 4(E; top)) is insufficient to adjust to new training data, while training biases and scales allows to approximate new class assignment (see figure 4(C; top)). Notice that a combination of all three types of trainable parameters (biases, scales and logits) frequently results in the best function approximation even if the initial state is chosen to be random (see figure 4(A)-(E); bottom row).

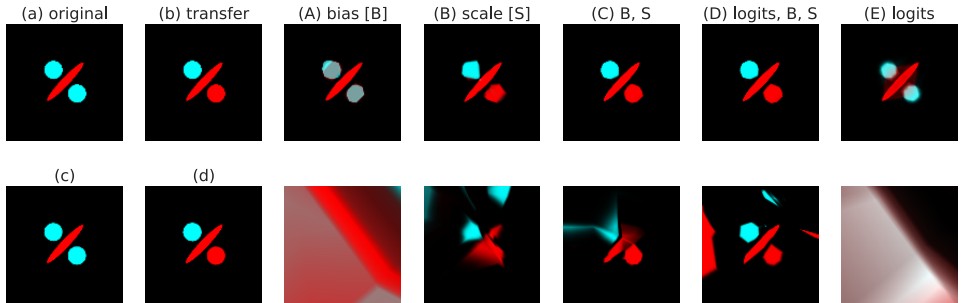

Figure 4: The neural network is first trained to approximate class assignment shown in (a) (with the corresponding learned outputs in (c)), network parameters are then fine-tuned to match new classes shown in (b). If all network parameters are trained, it is possible (d) to get a good approximation of the new class assignment. Outputs obtained by fine-tuning only a subset of parameters are shown in columns (A) through (E): functions fine-tuned from a pretrained state (c) are shown at the top row; functions trained from a random state (the same for all figures) are shown at the bottom. Each figure shows training with respect to a different parameter set: (A) biases; (B) scales; (C) biases and scales; (D) logits, biases and scales; (E) just logits.

Interestingly, poor performance of logit fine-tuning seen in figure 4(E) extends to higher embedding dimensions as well. Plots similar to those in figure 4, but generated for the model with the embedding dimension $m$ of 4 are shown in figure 5. In this case, we can see that the final layer fine-tuning is again insufficient to achieve successful transfer learning. As the embedding dimension goes higher, last layer fine-tuning eventually reaches acceptable results (see figure 6 showing results for $m = 8$).

The explanation behind poor logit fine-tuning results can be seen by plotting the embedding space of the original model with $m = 2$ (see figure 7(a)). Both circular regions are assigned the same embedding and the final layer is incapable of disentangling them. But it turns out that the same network could have learned a different embedding that would make last layer fine-tuning much more efficient. We show this by training the network on the classes shown in figure 7(b). This class assignment breaks the symmetry and the new learned embedding shown in figure 7(c) can now be used to adjust to new class assignments shown in figure 7(d), (e) and (f) by fine-tuning the final layer alone.

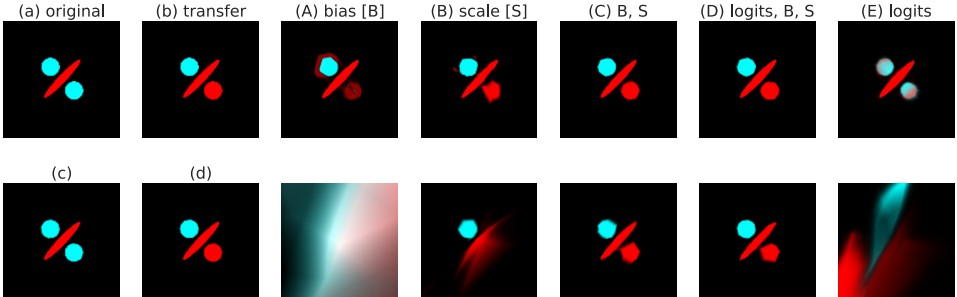

Figure 5: Plots similar to those shown in figure 4, but obtained for the embedding dimension of 4.

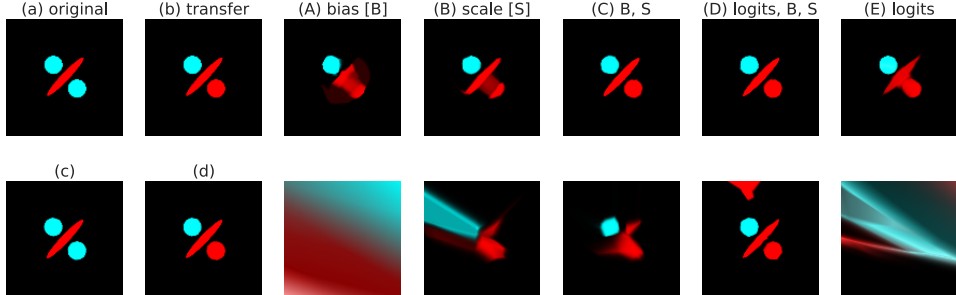

Figure 6: Plots similar to those shown in figure 4, but obtained for the embedding dimension of 8.

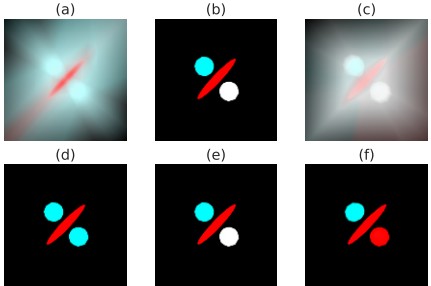

Figure 7: Original model trained to match class assignment shown in figure 4(a) results in the embedding shown in (a) that "folds" both circular regions together. After training the same model on different classes (b), the new embedding (c) allows one to fine-tune the last layer alone to obtain outputs shown in (d), (e) and (f).

## B    ADDITIONAL EXPERIMENTS

### B.1    ADJUSTING BATCH-NORMALIZATION STATISTICS

The results of Li et al. (2016) suggested that adjusting Batch Normalization statistics helps with domain adaption. Interestingly we found that it significantly worsens results for transfer learning, unless bias and scales are allows to learn. We find that fine-tuning on last layer with batch-norm statistics readjusted to keep activation space at mean 0/variance 1, makes the network to significantly under-perform compared to fine-tuning with frozen statistics. Even though adding *learned* bias/scales signifcanty outperforms logit-only based fine-tuning. We summarize our experiments in Table 6

Table 6: The effect of batch-norm statistics on logit-based fine-tuning for MobileNetV2

| Method | Flowers | Aircraft | Stanford Cars | Cifar100 |
|---|---|---|---|---|
| Last layer (logits) | 80.2 | 43.3 | 51.4 | 45.0 |
| Same as above + batchnorm statistics | 79.8 | 38.3 | 43.6 | 57.2 |
| Same as above + scales and biases | **86.9** | **65.6** | **75.9** | **74.9** |

### B.2    ACCURACY VS. LEARNING RATE

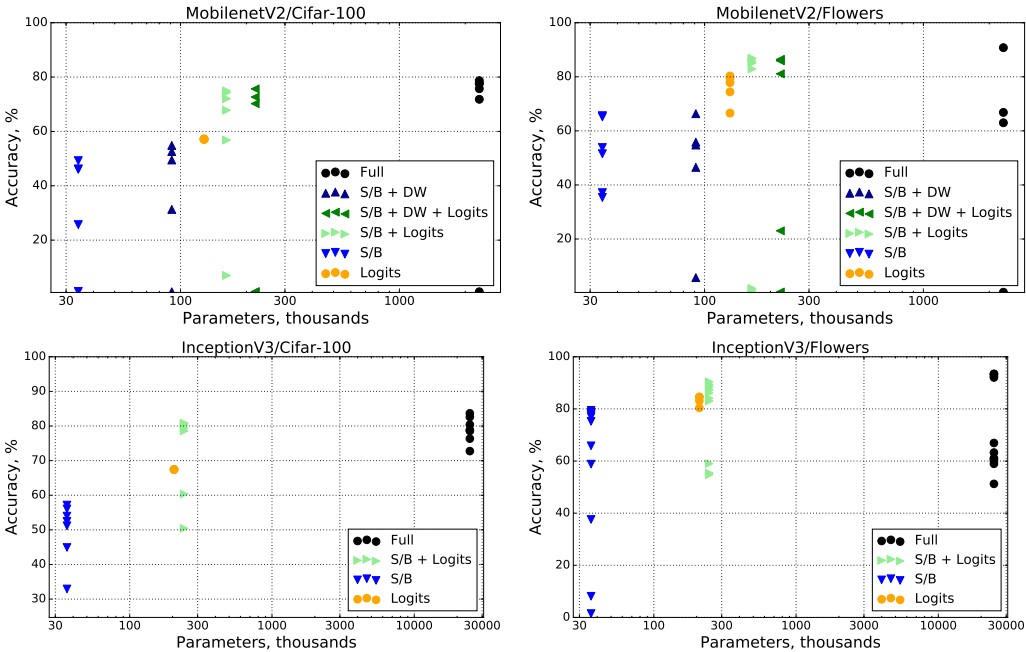

Figure 8: Performance of different fine-tuning approaches for Mobilenet V2 and Inception V3 for Cifar100 and Flowers. Best viewed in color.

## C    MODEL CASCADES

An application of domain adaptation using model patches is cost-efficient model cascades. We employ the algorithm from Streeter (2018) which takes several models (of varying costs) performing the same task, and determines a cascaded model with the same accuracy as the best task but lower average cost. Applying it to MobilenetV2 models on multiple resolutions that we trained via multi-task learning, we are able to lower the average cost of MobilenetV2 inference by 15.2%. Note that, in order to achieve this, we only need to store 5% more model parameters than for a single model.

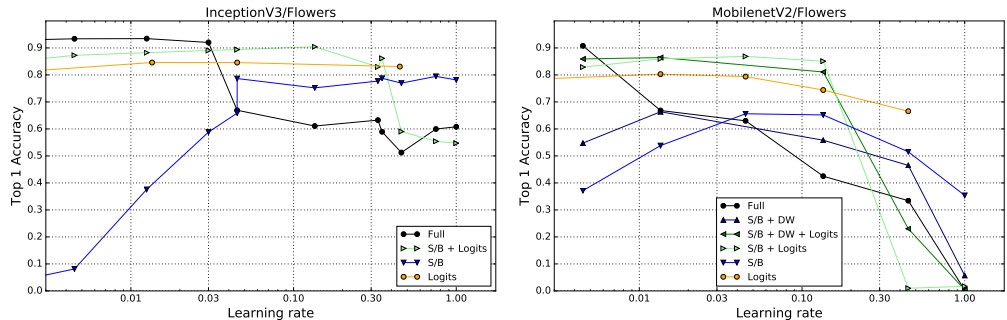

Figure 9: Final accuracy as a function of learning rate. Note how full fine-tuning requires learning rate to be small, while bias/scale tuning requires learning rate to be large enough.

## D    NOTE ABOUT TRAINING SPEED

Generally, we did not see a large variation in training speed. All fine-tuning approaches needed 50-200K steps depending on the learning rate and the training method. While different approaches definitely differ in the number of steps necessary for convergence, we find these changes to be comparable to changes in other hyperparameters such as learning rate.

