# OpenReview forum: "K for the Price of 1: Parameter-efficient Multi-task and Transfer Learning"
_ICLR.cc/2019/Conference_

### Official Review · AnonReviewer1 · 2018-10-21
**Interesting idea and fair evaluation. Accept with minor changes.**

**Rating:** 8
**Confidence:** 4

**Review:**

Summary: the paper introduces a new way of fine-tuning neural networks. Instead of re-training the whole model or fine-tuning the last few layers, the authors propose to fine-tune a small set of model patches that affect the network at different layers. The results show that this way of fine-tuning is superior to above mentioned typical ways either in accuracy or in the number of tuned parameters in three different settings: transfer learning, multi-task learning and domain adaptation.

Quality: the introduced way of fine-tuning is interesting alternative to the typical last layer re-training. I like that the authors present an intuition behind their approach and justify it by an illustrative example. The experiments are fair, assuming the authors explain the choice of hyper-parameters during the revision.

Clarity: in general the paper is well-written. The discussion of multi-task and domain adaptation parts can be improved though.

Originality: the contributions are novel to my best knowledge.

Significance: high, I believe the paper may facilitate a further developments in the area.

I ask the authors to address the following during the rebuttal stage:
* explain the choice of the hyper-parameters of RMSProp (paragraph under Table 1).
* fix Figure 3, it's impossible to read in the paper-printed version
* explain how the average number of parameters per model in computed in Tables 4 and 5. E.g. 700K params/model in the first column of Table 4 is misleading - I suppose the shared parameters are not taken into account. The same holds for 0 in the second column, etc.
* add a proper discussion for domain adaptation part. The simple "The results are shown in Table 5" is not enough.
* consider leaving the discussion of cost-efficient model cascades out. The presented details are too condensed and do not add value to the paper.
* explain how different resolutions are managed by the same model in the domain adaptation experiments.

---

> ### Author Response · Authors · 2018-11-17
> **Response to AnonReviewer1**
>
> We thank AnonReviewer1 for the review. Below are our responses inline.
>
> >> * explain the choice of the hyper-parameters of RMSProp (paragraph under Table 1).
>
> The hyper-parameters are the same as those in the standard setup for MobilenetV2 or InceptionV3. We have added a line in the experiments section mentioning this.
>
> >> * fix Figure 3, it's impossible to read in the paper-printed version
>
> The four subfigures are now split into two rows and are now hopefully easily readable.
>
> >> * explain how the average number of parameters per model in computed in Tables 4 and 5. E.g. 700K params/model in the first column of Table 4 is misleading - I suppose the shared parameters are not taken into account. The same holds for 0 in the second column, etc.
>
> Thank you for pointing this out. We had mistakenly only counted the non-shared parameters in the models, and forgot to include the last layer parameters in the second column. This has now been corrected to simply the total number of parameters trained.
>
> >> * add a proper discussion for domain adaptation part. The simple "The results are shown in Table 5" is not enough.
>
> Done.
>
> >> * consider leaving the discussion of cost-efficient model cascades out. The presented details are too condensed and do not add value to the paper.
>
> Makes sense. We moved these results to the appendix to be included in the full version.
>
> >> * explain how different resolutions are managed by the same model in the domain adaptation experiments.
>
> We added a line in the paper stating the images are brought to the right resolution using bilinear interpolation before passing as input to each model.

---

> > ### Comment · AnonReviewer1 · 2018-11-25
> > **Response to Authors**
> >
> > Thanks to the authors for their reply. I am satisfied with the current state of the paper and tend to keep my score.

---

### Official Review · AnonReviewer2 · 2018-11-02
**Inspiring thought, though lack of sufficient proofs**

**Rating:** 6
**Confidence:** 3

**Review:**

This paper explored the means of tuning the neural network models using less parameters. The authors evaluated the case where only the batch normalisation related parameters are fine tuned, along with the last layer, would generate competitive classification results, while using very few parameters comparing with fine tuning the whole network model. However, several questions are raised concerning the experiment design and analysis:
1. Only MobilenetV2 and InceptionV3 are evaluated as classification model, while other mainstream models such as ResNet, DenseNet are not included. Would it be very different regarding the conclusion of this paper?
2. It seems that the only effective manner is by fine tuning the parameters of both batch normalisation related and lasts layer, while fine tuning last layer seems to be having the main impact on the final result. In Table 4, authors do not even provide the results fine tuning last layer only.
3. The organisation of the paper and the order of illustration is a bit confusing. e.g. later sections are frequently referred in the earlier sections. Personally I would prefer a plain sequence than keep turning pages for confirmation.

---

> ### Author Response · Authors · 2018-11-17
> **Response to AnonReviewer2**
>
> We thank AnonReviewer2 for the review. Below is our detailed response.
>
> >> 1. Only MobilenetV2 and InceptionV3 are evaluated as classification model, while the residual connection based models such as ResNet, DenseNet are not included. Would it be very different regarding the conclusion of this paper?
>
> We experimented extensively with multiple tasks (classification, detection, multi-task learning) and datasets instead of trying more models for the same task, as we intended to test the effectiveness of our method in various situations. Further, MobileNetV2 has residual connections, which encouraged us to believe that the results on other residual connection based models would be similar.
>
> We ran experiments with ResNet and got similar results. For instance, transfer learning accuracy from ImageNet to Cars goes up from 61.4% (last layer fine-tuning) to 73% (S/B patch + last layer fine-tuning). From ImageNet to Aircraft, accuracy goes up from 51.8% (last layer) to 62.5% (S/B patch + last layer). In the interest of space, we did not think it added much to the experimental section of the paper.
>
> >> 2. It seems that the only effective manner is by fine tuning the parameters of both batch normalisation related and lasts layer, while fine tuning last layer seems to be having the main impact on the final result. In Table 4, authors do not even provide the results fine tuning last layer only.
>
> Fine-tuning the last layer is not always required. For instance, in domain adaptation (Sec 5.4), the model patch consists of only the batch normalization parameters, and the resulting accuracies match or exceed those of individually trained models.
>
> From Figure 3 and Table 4, we see that fine-tuning scales, biases (S/B) and depthwise (DW) along with last layer causes an average 50% relative improvement in accuracy over fine-tuning only the last layer while being only a small (4%) increase in terms of number of parameters over the last layer.
>
> When performing multi-task or transfer learning across different tasks (e.g. ImageNet → Places365), it becomes necessary to have different last layers as the output spaces are different. In Table 4, the second column corresponds to the case where only the last layer is separate for each task. We apologize if this was not clear - we have now updated Table 4 headers to explicitly reflect this fact.
>
> >> 3. The organisation of the paper and the order of illustration is a bit confusing.
>
> We will be happy to modify the paper if the reviewer elaborates on this point.

---

> > ### Comment · AnonReviewer2 · 2018-11-19
> > **Thanks for the response**
> >
> > Several changes have been made to my comments, thanks for pointing out the mistakes.

---

### Official Review · AnonReviewer3 · 2018-11-04
**Interesting results on transfer learning**

**Rating:** 7
**Confidence:** 5

**Review:**

The authors proposed an interesting method for parameter-efficient transfer learning and multi-task learning. The authors show that in transfer learning fine-tuning the last layer plus BN layers significantly improve the performance of only fine-tuning the last layer. The results are surprisingly good and the authors also did analysis on the relationship between embedding space and biases.

1. The memory benefit is obvious, it would be interesting to know the training speed compared to fine-tuning methods (both the last layer and the entire network)?
2. It seems that DW patch has limited effects compared to S/B patch. It would be nice to have some analysis of this aspect.

---

> ### Author Response · Authors · 2018-11-17
> **Response to AnonReviewer3**
>
> We thank AnonReviewer3 for the review. Below are our responses to specific comments.
>
> >> 1. The memory benefit is obvious, it would be interesting to know the training speed compared to fine-tuning methods (both the last layer and the entire network)?
>
> Generally, we did not see a large variation in training speed on the datasets that we tried. All fine-tuning approaches needed 50-200K steps depending on the learning rate and the training method. While different approaches definitely differ in the number of steps necessary for convergence, we find these changes to be comparable to changes in other hyper-parameters such as learning rate, and generally not providing a clear signal worth articulating in the paper.
>
> >> 2. It seems that DW patch has limited effects compared to S/B patch. It would be nice to have some analysis of this aspect.
>
> Yes, DW patch seems to be less powerful than S/B patch. Generally, DW patch resulted in about 5-10% percentage points lower accuracy than the S/B patch while having comparable number of parameters. However, it does add a lot of value when used in conjunction with S/B patch. For instance, from the top two figures in Figure 3, we see that fine-tuning the combination of DW and S/B patches (4% of the network parameters) closes the accuracy gap between S/B patch (1% of the network parameters) and fine-tuning the last layer (37% of the network parameters).
>
> If the reviewer thinks that adding the performance of DW only patch would be a useful addition to Figure 3, we are happy to do that. We had excluded it in the interest of not crowding the plots.

---

### Meta-Review · Area_Chair1 · 2018-12-16
**Simple and effective parameter efficient method for finetuning**

**Confidence:** 4
**Recommendation:** Accept (Poster)

**Metareview:**

Reviewers largely agree that the proposed method for finetuning the deep neural networks is interesting and empirical results clearly show the benefits over finetuning only the last layer. I recommend acceptance.